# The Impact of Metabolic Memory on Immune Profile in Young Patients with Uncomplicated Type 1 Diabetes

**DOI:** 10.3390/ijms25063190

**Published:** 2024-03-10

**Authors:** Jolanta Neubauer-Geryk, Melanie Wielicka, Małgorzata Myśliwiec, Katarzyna Zorena, Leszek Bieniaszewski

**Affiliations:** 1Clinical Physiology Unit, Medical Simulation Centre, Medical University of Gdańsk, 80-204 Gdańsk, Poland; melanie.wielicka@gmail.com (M.W.); lbien@gumed.edu.pl (L.B.); 2Department of Pediatrics, Diabetology and Endocrinology, Medical University of Gdańsk, 80-211 Gdańsk, Poland; mysliwiec@gumed.edu.pl; 3Department of Immunobiology and Environment Microbiology, Medical University of Gdańsk, 80-211 Gdańsk, Poland; katarzyna.zorena@gumed.edu.pl

**Keywords:** metabolic memory, anti-inflammatory cytokines, pro-inflammatory cytokines, type 1 diabetes mellitus, young diabetics, lipids

## Abstract

Metabolic memory refers to the long-term effects of achieving early glycemic control and the adverse implications of high blood glucose levels, including the development and progression of diabetes complications. Our study aimed to investigate whether the phenomenon of metabolic memory plays a role in the immune profile of young patients with uncomplicated type 1 diabetes (T1D). The study group included 67 patients with uncomplicated type 1 diabetes with a mean age of 15.1 ± 2.3 years and a minimum disease duration of 1.2 years. The control group consisted of 27 healthy children and adolescents with a mean age of 15.1 ± 2.3 years. Patients were divided into three groups according to their HbA_1c_ levels at the onset of T1D, and the average HbA_1c_ levels after one and two years of disease duration. The subgroup A1 had the lowest initial HbA_1c_ values, while the subgroup C had the highest initial HbA_1c_ values. Cytokine levels (including TNF-α, IL-35, IL-4, IL-10, IL-18, and IL-12) were measured in all study participants. Our data analysis showed that subgroup A1 was characterized by significantly higher levels of IL-35 and IL-10 compared to all other groups, and significantly higher levels of IL-4 compared to group B. Additionally, a comparative analysis of cytokine levels between the groups of diabetic patients and healthy controls demonstrated that subgroup A1 had significantly higher levels of anti-inflammatory cytokines. The lipid profile was also significantly better in subgroup A1 compared to all other patient groups. Based on our findings, it appears that an inflammatory process, characterized by an imbalance between the pro- and anti-inflammatory cytokines, is associated with poor glycemic control at the onset of diabetes and during the first year of disease duration. These findings also suggest that both metabolic memory and inflammation contribute to the abnormal lipid profile in patients with type 1 diabetes.

## 1. Introduction

The concept of metabolic memory refers to the long-term benefits of achieving early glycemic control as well as the negative impact of hyperglycemia, including the development and progression of complications in diabetic patients [1]. The Diabetes Control and Complications Trial showed that maintaining excellent glycemic control with intensive insulin therapy significantly reduced the risk of early microvascular complications of diabetes by an average of 6.5 years compared to conventional treatment [2,3]. In fact, the DCCT demonstrated that intensive insulin therapy is more effective than conventional treatment in patients with type 1 diabetes (T1D). These findings were further validated during the first four years of the EDIC study.

It has been suggested that as a result of metabolic memory, ongoing glycemic control later in the disease course is not the most important factor in preventing diabetic complications. Instead, HbA1c levels between two and three years prior confer the highest risk of developing retinopathy [4] or nephropathy [3].

Although some of the adverse effects of hyperglycemia exposure begin to diminish after a few years, others may persist for longer [5]. Monnier et al. explained this by the gradual accumulation and eventual degradation of advanced glycation end products [4,6,7].

Hyperglycemia-induced endothelial cell dysfunction develops early in the course of type 1 diabetes [8] and leads to decreased neoangiogenesis in response to tissue ischemia. This results in a delayed response to injuries and the development of micro- and macrovascular complications. There is evidence that both the number and function of endothelial progenitor cells are significantly decreased in patients with diabetes. Circulating endothelial progenitor cell levels have been shown to be reduced by up to 50% in diabetic patients compared to nondiabetic controls [9]. Studies have also demonstrated that rats with diabetes have reduced levels of circulating endothelial progenitor cells as a result of neuropathy, a phenomenon thought to precede the development of diabetic retinopathy [10,11,12].

Studies investigating experimental models of metabolic memory have indicated that epigenetic mechanisms are implicated in the adverse effects of hyperglycemia [13,14,15,16,17,18,19,20,21,22,23,24,25]. Epigenetic modifications may occur as a result of exposure to environmental factors, such as dietary changes. They can be a predisposing factor for the development of various disease processes, including diabetes [14]. These modifications have the ability to impact gene expression and phenotype without altering the DNA sequence [17,19,26,27,28,29].

The role of cytokines in the pathophysiology of T1D is complex and not fully understood. Recent studies suggest that cytokines play an important role in the pathogenesis of the disease [21,30,31,32]. Most of the studied inflammatory markers have been shown to change gradually over time [22].

TNF-α is a cytokine that plays a critical role in the inflammatory response [13,23,24,25,26,33]. TNF-α has been associated with diabetic complications [27]. In fact, TNF-α has been detected in the serum of children and young adults with type 1 diabetes and non-proliferative diabetic retinopathy [27,34,35,36,37]. One study found that plasma TNF-α levels were significantly higher in diabetic patients than in healthy subjects. Additionally, TNF-α levels positively correlated with age [31]. Available data on the relationship between TNF-α and glycemic control in type 1 diabetes are somewhat conflicting. Some studies have found no association [26], while others describe a beneficial effect of TNF-α in patients with long-standing disease [27].

Poor glycemic control in diabetes results in increased oxidative stress [31,38] and increased protein glycation. This has been linked to cytokine activity, including TNF-α, interleukin-6 (IL-6), and interleukin-12 (IL-12). These cytokines have been implicated in the development of diabetes and the associated microvascular complications [39,40,41]. Cytokines of the IL-12 family, namely IL-12, IL-27, IL-23, IL-35, and IL-39, play a crucial role in immunoregulation [36,42,43,44,45,46,47]. Diabetes complications, such as delayed wound healing, retinopathy, atherosclerosis [36], and neuropathy [37], are mediated by the IL-12 family of cytokines. Abnormal levels may worsen glucose variability and increase the risk of diabetic complications [48]. IL-35 levels have also been found to have a significant negative correlation with high HbA1c levels. This cytokine may play a crucial role in reducing inflammation and improve insulin sensitivity in T1D. In contrast, IL-18 had been shown to have a positive correlation with HbA1c levels [49]. Harms et al. demonstrated a positive correlation between IL-18, IL-18 BP, and HbA1c levels in patients with T1D. This suggests a potential association between IL-18 and glycemic control in these patients [38].

IL-10 gene expression was found to be significantly increased in patients with ketoacidosis and has been shown to have a positive correlation with HbA1c levels. In addition, studies have demonstrated a negative correlation between IL-10 expression and patient age at the time of diabetes diagnosis [31]. This cytokine is an important regulator of immunity during infection, playing a key role in reducing or terminating inflammation and resulting in host protection [50,51].

It has been shown that IL-4 has a protective effect against pancreatic beta-cell loss in type 1 diabetes and that IL-4 receptors in pancreatic beta cells are functionally competent [52]. IL-4, IL-10, and IL-13 cytokines have been proposed to activate the humoral immune response by stimulating B cells to release autoantibodies against islet cells and GAD molecules. Serum levels of cytokines such as IFN-γ, TNF-α, IL-6, IL-1β, IL-4, and IL-10 were significantly higher in patients with T1D compared to healthy individuals [41,49,53]. These findings suggest that hyperglycemia may upregulate inflammatory pathways and increase pro-inflammatory cytokine secretion in patients with T1D [54].

In type 1 diabetics, poor glycemic control is associated with elevated LDL cholesterol levels, which may be partly explained by the catabolic effect of insulin on LDL cholesterol. Insulin affects lipoprotein metabolism in several ways: it increases lipoprotein lipase activity and inhibits the production of VLDL particles in the liver. Thus, poor glycemic control in type 1 diabetes is associated with high levels of atherogenic triglyceride-rich lipoproteins and cholesterol-rich LDL particles [55,56]. Moreover, high levels of HbA1c in diabetic patients at the time of diagnosis have been associated with an increased incidence of late major cardiovascular events [57,58].

Type 1 diabetes is associated not only with quantitative differences in lipoprotein levels compared with controls but also with qualitative changes, such as increased VLDL cholesterol and increased triglycerides in the HDL fraction. Unfortunately, these qualitative changes are not completely reversible even with the optimization of glycemic control [59]. There is also emerging evidence that impaired lipid metabolism begins one year before seroconversion in children with T1D [45].

The aim of this study was to investigate whether concentrations of pro- and anti- inflammatory cytokines and lipid profile abnormalities in young patients with uncomplicated type 1 diabetes could be associated with the metabolic memory assessed by HbA1c levels in the very first years of disease.

## 2. Results

### 2.1. Characteristics of Studied Subgroups

Initially, 67 patients with type 1 diabetes were included in this study (Figure 1). The patients were divided into three groups based on their HbA_1c_ levels at the time of disease diagnosis, and the mean levels obtained during the first and second year after diagnosis (Figure 2). The resulting groups of patients A, B, and C did not differ in chronological age and age at the onset of the disease. However, group A was significantly different from group C regarding diabetes duration (*p* = 0.004). For this reason, the cluster has been divided into two subgroups, A1 and A2, according to median diabetes duration (9.87 years) (Figure 3). The demographic characteristics of the A1 subgroup did not differ from those of the B and C subgroups (Table 1).

The subgroups also did not differ significantly with regard to patient gender, BMI, duration of insulin pump use or insulin dose, or the number of mild and severe hypoglycemia episodes. Additionally, the studied subgroups did not differ regarding thyroid hormones, creatinine levels, albuminuria, or CRP (Table 2).

It should be noted that the subgroups did not differ in mean HbA_1c_ levels obtained during the second year after diagnosis. However, a comparison of the current HbA_1c_ concentration showed a significant difference between subgroups B and C (*p* = 0.049). There was no significant difference between HbA_1c_ levels at disease onset and during the first year after diagnosis between girls and boys in any of the groups.

### 2.2. Cytokines Concentration

We compared the concentrations of the studied pro-inflammatory and anti-inflammatory cytokines between subgroups A1, B, and C and the controls (group H) (Table 2).

Comparative analysis of the cytokine levels between the A1, B, C, and H groups revealed statistically significant differences (Figure 4). The IL-35 levels were significantly higher in subgroup A1 compared with subgroup B (*p* < 0.001), C (*p* < 0.001) and the control (*p* < 0.001). The IL-4 levels were also significantly higher in subgroup A1 than in subgroup B (*p* = 0.03) and the control (*p* < 0.001). The IL-10 levels were significantly higher in subgroup A1 than in B (*p* = 0.01) and C (*p* < 0.001) subgroups. Moreover, the IL-10 levels were significantly higher in subgroup H than in subgroups B (*p* < 0.001) and C (*p* < 0.001), whereas subgroups A1 and H and B and C did not show statistically significant differences in IL-10 levels. The IL-18 levels were comparable in the diabetic subgroups, whereas they were statistically higher in these subgroups compared to the controls (*p* < 0.001). The TNF–α levels were significantly higher in subgroup C compared to subgroups A1 (*p* < 0.001) and B (*p* = 0.003).

The levels of TNF–α were significantly higher in the subgroups of patients with diabetes when compared to the control group (for A1, *p* = 0.002; for B and C, *p* < 0.001) (Table 2).

### 2.3. Laboratory Examination in Diabetic Subgroups and Healthy Control

When comparing the subgroups of patients with diabetes and the control group, the total cholesterol level was statistically significantly higher in subgroup C compared with subgroup A1 (*p* = 0.009) and the control group (*p* < 0.001). The LDL cholesterol level was statistically significantly higher in all subgroups of patients with diabetes compared to the control group (*p* < 0.001). In addition, the LDL cholesterol levels were significantly higher in subgroup C compared to subgroup A1 (*p* = 0.006). On the other hand, the HDL cholesterol levels were statistically significantly higher in subgroups A1 (*p* = 0.005) and B (*p* = 0.04) compared to the control group (Table 2, Figure 5).

### 2.4. Characteristics of Subgroups A1 and A2

The duration of disease in group A was found to be significantly different from group C. Thus, cluster A was additionally divided into two subgroups, A1 and A2, based on the median of diabetes duration. The subgroups did not differ with regard to HbA_1c_ at diabetes onset or after the first and second years. They also do not differ from each other in their current HbA_1c_ levels (Table 3).

Subgroup A1 had significantly higher levels of IL-35 and IL-4, as well as IL-12, and had a significantly shorter diabetes duration than subgroup A2 (Table 4). There was no difference in TNF—α levels between the subgroups (Figure 6). Given the differences in disease duration between the subgroups, the cytokine levels may be influenced by the duration of diabetes. A shorter duration was associated with anti-inflammatory cytokine predominance. On the other hand, in the subgroup with longer duration, the effect of the metabolic memory slowly fades away. The duration of diabetes in this A2 subgroup is almost more than 10 years (12.3 ± 1.8 years).

The prevalence of autoimmune thyroiditis and celiac disease in patient subgroups A1 and A2 were not statistically different (Table 3).

## 3. Discussion

The concept of metabolic memory was demonstrated in an independent analysis of DCCT data [60]. This results in the development of a mathematical model to investigate metabolic memory and evaluate its “shape” over time. The authors found that the effects of metabolic memory last up to 8 years. Subsequent analysis by DCCT/EDIC found that a period of hyperglycemia or hypoglycemia both have either positive or negative long-term effects, respectively, on the risk of micro- and macrovascular complications. The effects of metabolic memory can persist for up to 10 years before diminishing [3]. Several studies have previously analyzed the impact of metabolic memory on the severity of inflammation. Cé et al. [46], in a flow-mediated dilation study, showed that endothelium dysfunction is common in young patients with T1D. It is associated with a disease duration of fewer than five years and average levels of HbA_1c_ during the second year after diagnosis. Nathan et al., on behalf of the DCCT/EDIC study, stated that intensive glycemic control is associated with a reduced risk of microvascular complications [61]. Gubitosi-Klug et al. have demonstrated that this effect is sustained for up to 18 years of follow-up [47]. Type 1 diabetes mellitus is caused by the immune destruction of insulin-producing pancreatic β-cells. Certain cytokines released by intratumoral immune cells, such as IL-1B and TNF-a, contribute to the development of T1D by giving rise to B cell dysfunction [50,51]. Plasma cytokine levels in T1D may reflect pancreatic injury and serve as biomarkers of disease activity.

To the best of our knowledge, the association of metabolic memory with pro- and anti-inflammatory cytokine levels has not been studied. Our study was conducted in young patients with uncomplicated type 1 diabetes. We analyzed the levels of six cytokines in patients who differed in their HbA_1c_ levels at the onset of the disease and after the first year of disease duration. In particular, we were interested in analyzing whether the interplay between the cytokines studied changes depending on the level of glucose impairment at disease onset. The pro- and anti-inflammatory cytokines in diabetes mellitus were investigated by many researchers. A meta-analysis of serum TNF-α levels in patients with type 1 diabetes showed that patients with T1D had significantly higher serum TNF-α levels. TNF-α was significantly associated with patient age, disease duration, and ethnicity [62]. This is in partial agreement with the results of our study. The levels of TNF-alpha were significantly higher in the young diabetic patients in the study compared to the control group. Some studies have also found an association between cytokine levels and the duration of type 1 diabetes [22,62,63]. In the Norwegian Atherosclerosis and Childhood Diabetes prospective study, the analysis of TNF-α and IL-18 levels, among others, showed that the early, low-grade inflammation present in young people with T1D five years after diagnosis persists through ten years of disease, with moderate changes in most inflammatory markers over time [22]. To avoid the potential impact of diabetes duration on the final between-group comparison, we divided subgroup A into two subgroups (A1 and A2) according to median diabetes duration. This allowed us to create study subgroups that did not differ by age, diabetes onset, and diabetes duration.

A comparative analysis of the two subgroups, A1 and A2, demonstrated that with similar initial glycemic control, i.e., the same metabolic memory, as well as similar age, the cytokine system is also significantly influenced by diabetes duration and probably the age of diabetes onset in the A2 subgroup before the age of 10 years. Along these lines, Rawshani et al. showed that the risk of Atherosclerotic Cardiovascular Disease in patients with type 1 diabetes with disease onset before the age of 10 was approximately three times higher than in those who had onset between the ages of 26 and 30 years [64], even after adjustment for disease duration.

The comparison of cytokine levels between the diabetic subgroups and controls showed that subgroup A1 had higher levels of anti-inflammatory cytokines than the controls. Subgroup A1 was characterized by significantly higher levels of IL-35 and IL-10 than the other subgroups of patients with diabetes, and statistically higher IL-4 when compared to subgroup B. Our study shows that the effect of metabolic memory fades away with the increasing duration of diabetes. The A2 subgroup with a longer diabetes duration did not differ in TNF-α levels, while the IL-35 and IL-4 levels were significantly lower compared to the A1 subgroup. The high levels of IL-35 in the subgroup A1, which are significantly higher compared to the healthy control, could be a consequence of the significantly lower HbA_1c_ in the first year of the disease. This could represent a mechanism against inflammation that disappears after the first 10 years of disease duration. Our results differ from those published by Young Chao Qiao et al. in a meta-analysis. They have shown that serum TNF-α levels are significantly elevated in T1D patients of all ages and disease duration [62]. On the other hand, our results regarding the IL-12 levels are in line with those reported by Kowalewska et al. The authors have demonstrated that serum IL-12 levels decreased by 18.1 pg/mL per year and decrease by 52.9 pg/mL for every 1% increase in HbA_1c_ [63].

Although the A1 and A2 subgroups of patients with diabetes were statistically different in terms of the duration of the disease, the current HbA_1c_ levels were comparable between the subgroups. Our data are in agreement with the findings of Al-Dubayee et al. [49] while others have shown that a longer duration of diabetes is associated with higher HbA_1c_ levels [65,66]. In our study, we have shown (Table 2) that there were no differences in the levels of pro-inflammatory cytokines, IL-12 and IL-18, between all subgroups. However, there were significant differences in the levels of anti-inflammatory cytokines (IL-35, IL-4, and IL-10) and pro-inflammatory cytokines (TNF-a). Subgroup C, which had very poor glycemic control, had the highest levels of TNF-a and significantly lower levels of IL-35 and IL-10 compared to the other subgroups. This finding is in line with a study by Overgaard et al., which has demonstrated that in pediatric patients with type 1 diabetes, TNF-α correlates with residual beta-cell function and may serve as a prognostic marker of disease progression [67]. Other studies have also shown elevated levels of TNF-α in patients with T1D [26,27,33,49]. Our study results demonstrated that glycemic control in the first year of disease (HbA_1c_ onset, HbA_1c_ 1st year) has a significant impact on inflammation even years later.

In our group of patients, however, cytokine levels did not depend on the current level of glycemic control. We have noticed only a marginally significant correlation of current HbA_1c_ between subgroups B and C, whereas current HbA_1c_ did not differ significantly between subgroups A1 and B and A1 and C. This finding emphasizes that metabolic memory related to early glycemic control, rather than current HbA_1c_, affects current cytokine levels.

Prior studies have shown an ambiguous relationship between present HbA_1c_ levels and cytokine levels in diabetic patients. Some authors have shown a significant positive correlation between TNF-α and HbA_1c_ [65]. On the other hand, some studies have demonstrated a significant negative correlation between the levels of IL-12 [63], IL-35 [49], and HbA_1c_ levels. Similarly to our results, Al-Dubayee et al. found no significant relationship between higher levels of HbA_1c_ and TNF-α IL-4 or IL-10 and the HbA_1c_ level [49].

The results of our study showed that higher levels of total and LDL cholesterol were found in the subgroup with poor glycemic control in the past. This subgroup was also characterized by higher TNF-α levels, which have been shown to contribute to early atherosclerosis by increasing the subendothelial retention of LDL cholesterol in the vascular walls [66]. It is well known that LDL cholesterol is the most abundant atherogenic lipoprotein in plasma. Endothelial dysfunction is an early stage of atherosclerosis and is characterized by, among other things, increased oxidation of LDL cholesterol. In our study, the A1 subgroup, which was characterized by the best early glycemic control among all groups of patients with diabetes, did not differ from group C with the worst glycemic control. Interestingly, lower levels of total and LDL cholesterol and higher levels of HDL cholesterol were observed in the A1 subgroup compared to group C. This indicates that worse glycemic control at the onset of the disease may imply the disturbation in lipid control. Considering that our study was performed in patients with uncomplicated T1D, we can refer to the results obtained in other published studies.

The DCCT study [55] has shown a strong association between higher levels of current HbA_1c_ and elevated levels of triglycerides and LDL cholesterol. Similar results have also been obtained by other researchers [56,68]. Similarly, analysis of the SEARCH study showed that poor glycemic control over time is a risk factor for the progression of dyslipidemia [69]. However, Prado et al. did not show this association in a small group of children and adolescents with T1D [70]. Verges has pointed out that in T1D with good glycemic control, triglycerides and LDL cholesterol are normal or decreased, while HDL cholesterol is normal or increased [61].

In our present study, in the group of patients aged 8 to 18 years, poor glycemic control during the initial period of the disease—the onset of the disease and the first year—is associated with a predominance of pro-inflammatory cytokines. This is independent of the current HbA_1c_ levels, duration of the disease, age at onset, and chronological age of the patients studied. We hypothesized that our findings my reflect the prognostic importance of so-called metabolic memory.

The fact that the diabetic children and adolescents included in this study did not have any microangiopathic complications is at the same time a limitation and a strength of our study. This helped us demonstrate the imbalance between the pro- and anti-inflammatory cytokines and lipid profiles. It would be worthwhile to compare more recent cytokine levels with those at the time of diagnosis. This would provide an additional source of information about changes in the severity of inflammation with disease progression and allow us to analyze such changes in relationship to early glycemic control. Unfortunately, we have no such data at this time.

## 4. Materials and Methods

### 4.1. Study Group

The study group included 67 patients diagnosed with type 1 diabetes, 37 girls and 30 boys (Table 1), aged between 8.4 to 18 years with diabetes of more than 1.2 years who met the type 1 diabetes diagnostic criteria according to the International Society of Child and Adolescent Diabetes [71]. Patients were divided into three groups, A1, B, and C, based on their HbA_1c_ levels at the time of diagnosis and average HbA_1c_ levels obtained during the first and second year after diagnosis., with a mean age of 16.1 ± 2.3, 14.8 ± 2.2, and 14.9 ± 1.9 years, respectively. The mean age at disease onset for each group was 6.5 ± 3.6, 7.8 ± 3.4, and 9.0 ± 3.7 years, and mean diabetes duration 9.6 ± 3.5, 7.0 ± 3.9, and 5.9 ± 2.9 years, respectively. Exclusion criteria included diabetic ketoacidosis at the time of enrollment, ongoing infection, uncontrolled celiac disease, chronic kidney disease, hypothyroidism, or other endocrine disorders. Patients having experienced severe hypoglycemia in the previous month were excluded. All the participants included in the healthy control group were young people with no history of chronic disease who did not take any medications.

Patients were enrolled from 2014 to 2018 from the Department of Pediatrics, Diabetology, and Endocrinology of the Medical University of Gdańsk.

None of the patients had any form of microvascular disease such as retinopathy, nephropathy, or neuropathy according to medical history, physical examination, and biochemical analysis.

The patients treated with statin therapy were excluded from analysis.

Severe hypoglycemia was defined as an incident of blood glucose levels < 54 mg/dL within one year of the survey, but no more than one month prior to the survey, that required assistance from another person. Mild hypoglycemia was defined as an incident within one month prior to the survey with no need for assistance [57].

Mean HbA_1c_ levels during the first two years of disease were obtained from patients’ medical records from the Department of Pediatrics, Diabetology, and Endocrinology.

All patients in the study groups were euthyroid during the study period. The examinations were conducted while the subjects were fasting, within a specific time frame, from 8 a.m. to 1 p.m. The study protocol included laboratory tests.

The control group consisted of 27 healthy children and adolescents with a mean age of 15.1 ± 2.3 years (Table 1), recruited from schools or patients referred to the Diabetology Clinic for suspected endocrine disorders that were eventually ruled out.

The study was performed according to the ethical standards of the Ethical Committee of the Medical University of Gdańsk and the Declaration of Helsinki of 1964, as amended, or comparable ethical standards. The Medical University of Gdańsk Ethics Committee approved the study protocol (NKBBN/277/2014; NKBBN/277-512/2016). Informed consent was obtained from all participants. Parents also consented and participated with their children.

### 4.2. Analyses

Blood samples were collected between 7 and 9 a.m. after an overnight fast. Sera were separated from venous blood within 30 min and stored frozen at −80 °C for up to three months before analysis. The same blood sample was used for all measurements.

HbA_1c_ was measured by an immunoturbidometric method using the Unimate 3 set (Hoffmann-La Roche AG, Basel, Switzerland) with a normal range of values from 3.0 to 6.0%. An enzymatic test (Roche Diagnostics GmbH, Mannheim, Germany) was used to measure fasting glucose. The level of C-reactive protein was measured by an immunochemical system (Beckman Instr. Inc., Galway, Ireland). The levels of total cholesterol, HDL cholesterol, LDL cholesterol, and triglycerides were measured using Cormay enzymatic kits (Cormay, Lublin, Poland). Urinary albumin excretion was measured by an immunoturbidometric assay. Tina-quant was used (Boehringer Mannheim GmbH, Mannheim, Germany). Serum creatinine levels were measured using the CREA assay system (Boehringer Mannheim GmbH). Serum concentrations of IL-4, IL-10, IL-18, IL-12, IL-35, and TNF-α were measured by ELISA according to the manufacturer’s protocol. Serum levels of IL-4, IL-10, and IL-18 were measured by immunoenzymatic ELISA (Quantikine High Sensitivity Human from R&D Systems, Minneapolis, MI, USA) according to the manufacturer’s protocol. Minimum detectable concentrations were determined by the manufacturer as 10 pg/mL, 0.5 pg/mL, and 5.15 pg/mL, respectively. Intraassay (2.7%, 6.6%, and 2.9%) and interassay (7.4%, 8.1%, and 8.4%) precision’s performances of the assays were determined, respectively, for IL-4, IL-10, and IL-18. Serum concentrations of TNFα and IL-12 were measured by ELISA (Quantikine High Sensitivity Human from R&D Systems, Minneapolis, MI, USA) according to the manufacturer’s protocol. The intra- and interassay coefficients for TNFα and IL-12 were 6.2%, 2.6%, 2.5%, and 7.6%, respectively. Human IL-35 was measured by ELISA (Thermo Fisher Scientific, Inc., Waltham, MA, USA). Interassay CV < 10% and intraassay CV < 10%. Sensitivity: 9.38 pg/mL. The absorbance of IL-4, IL-10, IL-18, IL-35, IL-12, and TNF-α was read at 450 nm on a CHROMATE 4300 automated plate reader (Awareness Technology, Inc., Palm City, FL, USA). The reference curve was generated according to the manufacturer’s recommendations.

Glycated hemoglobin in the 1st and 2nd year of disease duration was determined as the average of 4 measurements taken every 3 months in the same laboratory.

### 4.3. Statistics

All statistical analysis of the data obtained was performed using SAS^®^ OnDemand for Academics, SAS Institute Inc., SAS Campus Drive, Cary, NC 27513, USA.

The clusters were calculated by the SAS FASTCLUS procedure. The distribution of the variables was assessed using the Shapiro–Wilk test. For comparison of non-normally distributed variables, the non-parametric ANOVA Kruskal–Wallis with a Tukey post hoc test was used. The chi-square test was used to compare gender proportions and hypoglycemic episodes. A significance level of *p* < 0.05 was considered statistically significant.

## 5. Conclusions

In summary, our data allow for indicating that a long-term inflammatory process, characterized by an imbalance between pro- and anti-inflammatory cytokines, is associated with poor glycemic control at diabetes onset and during the first year of disease duration. These findings suggest that the metabolic memory formed as a result of hyperglycemia in the early stages of T1D may modulate inflammatory pathways and lead to significant changes in the balance of pro- and anti-inflammatory cytokines. Both metabolic memory and the inflammation caused by a predominance of pro-inflammatory cytokines may be responsible for an abnormal lipid profile.

## Figures and Tables

**Figure 1 ijms-25-03190-f001:**
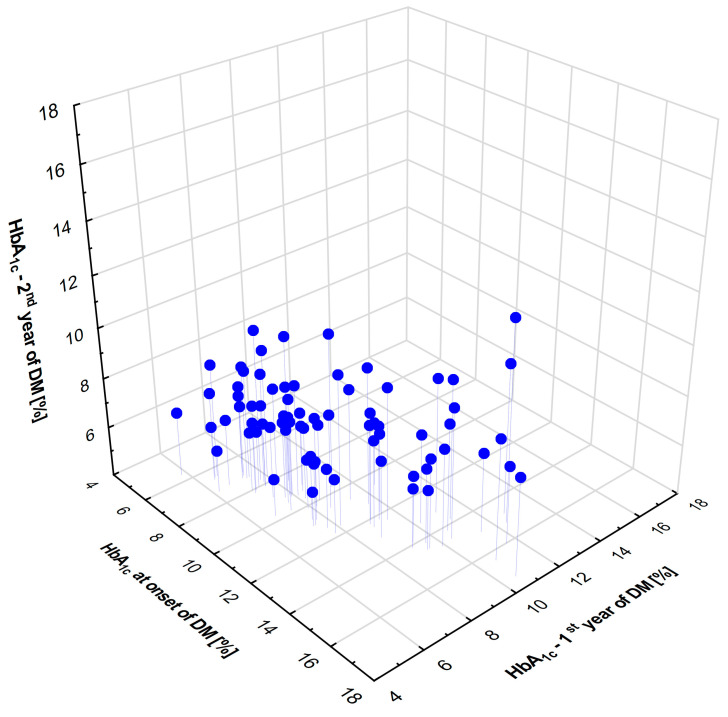
Characteristics of study groups divided based on HbA_1c_ levels at disease onset, and mean HbA_1c_ values obtained after first and second year following diagnosis.

**Figure 2 ijms-25-03190-f002:**
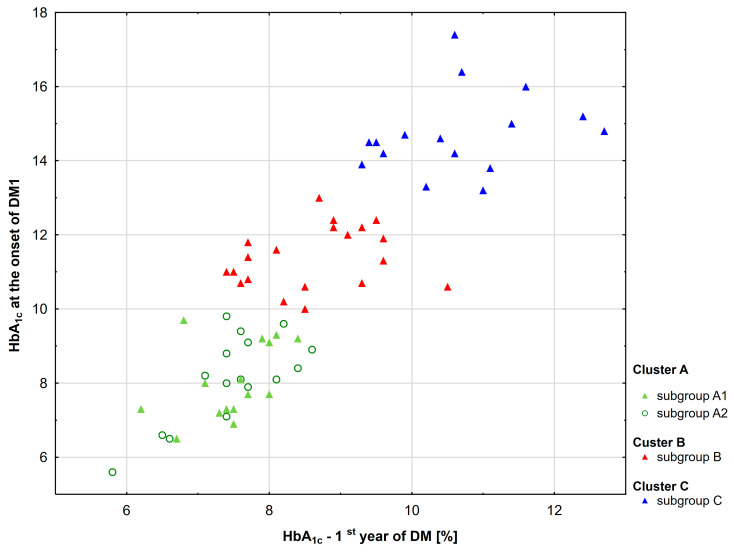
Patient groups based on HbA_1c_ levels at disease onset, and mean HbA_1c_ values obtained after first and second year following diagnosis. The triangles and circles indicate median values.

**Figure 3 ijms-25-03190-f003:**
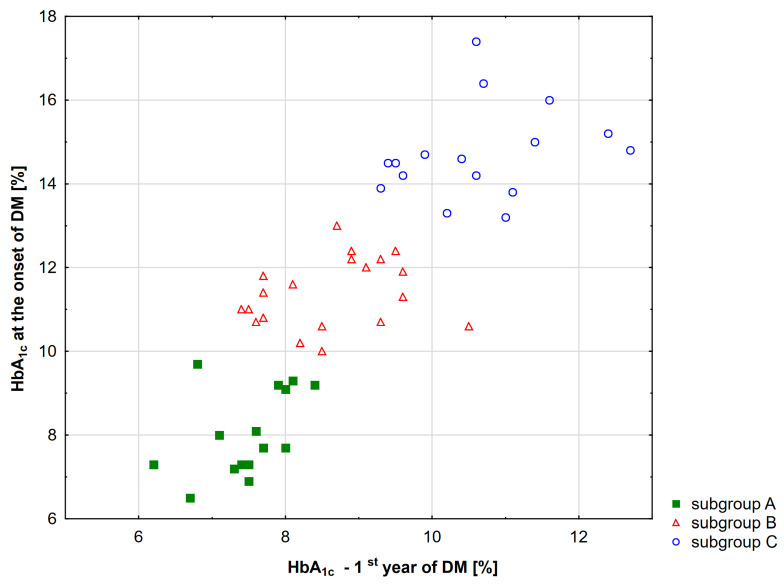
Demographic data in subgroups of diabetic patients in each group.

**Figure 4 ijms-25-03190-f004:**
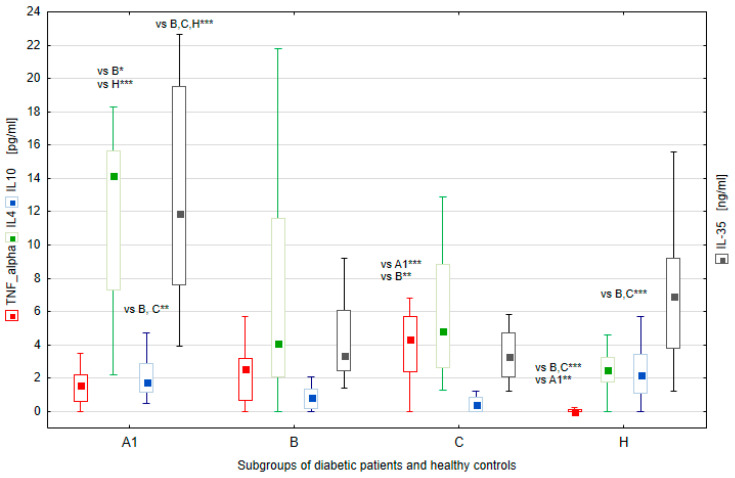
Differences in cytokine levels between patient groups (A1, B, and C) and healthy controls (H). Significant results are marked with * (*p* < 0.05), ** (*p* < 0.01), or *** (*p* < 0.001). Data are presented in box plots (median and range).

**Figure 5 ijms-25-03190-f005:**
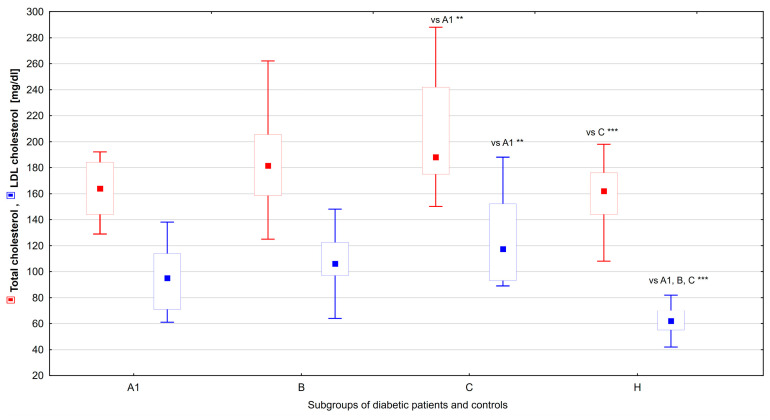
Differences in total and LDL cholesterol levels between groups of diabetic patients (A1, B, and C) and healthy controls (H). Significant results are marked with ** (*p* < 0.01), or *** (*p* < 0.001). Data are presented in box plots (median and range).

**Figure 6 ijms-25-03190-f006:**
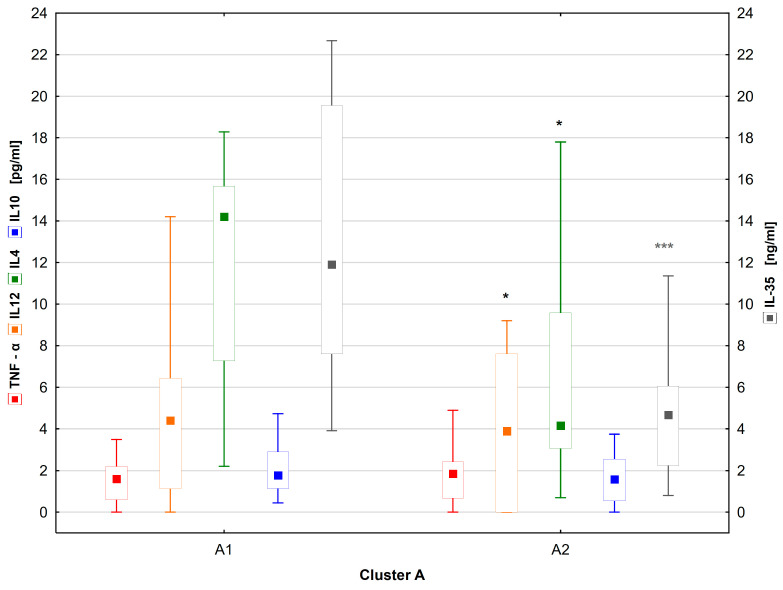
Comparison of individual cytokine levels between subgroups A1 and A2. Significant results are marked with * (*p* < 0.05), or *** (*p* < 0.001). Data are presented in box plots (median and range).

**Table 1 ijms-25-03190-t001:** Characteristics and comparison of the study in the whole group of patients with diabetes as well as in diabetic patients (A1, B, C) and healthy control (H). Data are presented as median (range)/mean values ± SD.

Characteristics	Healthy Control (H) n = 27	Diabetic Patients				*p*	Post Hoc Comparison					
			Subgroups				Between T1D Subgroups			Between T1D Subgroups and Control		
		Total n = 67	A1 n = 16	B n = 20	C n = 16		A1 vs. B	A1 vs. C	B vs. C	A1 vs. H	B vs. H	C vs. H
Males, n (%)	13 (48)	30 (45)	4 (25)	12 (60)	5 (31)	0.17						
BMI [kg/m^2^]	18.5 (15.6–26)	20.4 (14.5–29.7)	20.5 (15.3–26)	19.9 (14.5–27.1)	20.9 (15.0–29.7)	0.80						
Age [years]	14 (11–20)/ 15.1 ± 2.3	15.2 (8.4–18)/ 15.1 ± 2.3	17.1 (8.4–18)/ 16.1 ± 2.3	14.4 (10.9–17.8)/ 14.8 ± 2.2	15.2 (11.2–18)/ 14.9 ± 1.9	0.60						
Onset of diabetes [age]	na	9.1 (2.1–13.6)/ 8.5 ± 3.4	6 (1.2–13.5)/ 6.5 ± 3.6	8.2 (2.2–13.6)/ 7.8 ± 3.4	9.8 (2.1–13.6)/ 9.0 ± 3.7	0.50						
Diabetes duration [years]	na	6.9 (1.2–12.9)/ 6.6 ± 3.2	10.9 (1.2–15.9)/ 9.6 ± 3.5	6.9 (1.8–12.9)/ 7.0 ± 3.9	5.5 (1.7–10.6)/ 5.9 ± 2.9	0.53						
Insulin dose units/24 h	na	46.5 (20–100)	40 (21–70)	42.5 (20–90)	47.5 (25–70)	0.73						
Insulin dose units/kg	na	0.8 (0.4–1.4)	0.7 (0.4–1.0)	0.8 (0.5–1.2)	0.8 (0.5–1.2)	0.73						
Treatment with pump [%]	na	60 (0–100)	78 (0–100)	60 (0–100)	0 (0–100)	0.28						
HbA_1c_ at onset of T1D [%]	na	11.4 (6.5–17.4)	8.1 (5.6–9.8)	11.4 (10.0–13.0)	14.6 (13.2–17.4)	***	***	***	***			
Mean HbA_1c_ 1st year of T1D [%]	na	8.7 (6.2–12.7)	7.5 (5.8–8.6)	8.6 (7.4–10.5)	10.6 (9.3–12.7)	***	**	***	***			
Mean HbA_1c_ 2nd year of T1D [%]	na	7.6 (5.4–12)	7.1 (5.5–10.1)	7.4 (5.4–10)	7.9 (6.4–12)	0.15						
HbA_1c_ current [%]	4.7 (4–5.7)	7.8 (5.9–13.4)	8.2 (5.9–11.6)	7.4 (6.3–11.0)	8.3 (6.8–13.4)	*	0.99	0.09	*	na	na	na
Episodes of mild hypoglycemia [N/last month]	na	10 (0–30)	10 (0–20)	10 (4–30)	5 (1–20)	0.20						
Episodes of severe hypoglycemia [N/last year]	na	0 (0–2)	0 (0–1)	0 (0–2)	0 (0–1)	0.94						
Autoimmune thyroiditis, n [%]	na	15 (22)	5 (31)	4 (20)	2 (12.5)	0.37						
Celiac disease, n [%]	na	14 (21)	4 (25)	4 (20)	2 (12.5)	0.59						

Significant results are marked with * (*p* < 0.05), ** (*p* < 0.01), or *** (*p* < 0.001). na- not applicable Patient subgroups A1, B, and C were not statistically different in prevalence of autoimmune thyroiditis and celiac disease (Table 1).

**Table 2 ijms-25-03190-t002:** Results and comparison of laboratory results in all diabetic patients, diabetic patients divided into subgroups (A1, B, and C), and healthy controls (H). Data are presented as median (range).

Characteristics	Healthy Control (H) n = 27	Diabetic Patients				*p*	Post Hoc Comparison					
			Subgroups				Between T1D Subgroups			Between T1D Subgroups and Control		
		Total n = 67	A1 n = 16	B n = 20	C n = 16		A1 vs. B	A1 vs. C	B vs. C	A1 vs. H	B vs. H	C vs. H
CRP [mg/L]	0.32 (0.02–1.28)	0.6 (0.1–4.9)	0.6 (0.2–4.8)	0.5 (0.1–3.7)	0.8 (0.1–4.9)	0.07						
Serum creatinine [mg/dL]	0.69 (0.45–0.99)	0.65 (0.45–0.88)	0.71 (0.45–0.86)	0.64 (0.53 –0.88)	0.59 (0.5–0.81)	0.14						
Albuminuria [mg/dL]	Not tested	7 (3–41)	14 (3–33)	6 (3–41)	7 (3–35)	***	0.68	0.5	0.98			
Total cholesterol [mg/dL]	162 (108–198)	178 (125–288)	164 (129–248)	182 (125–262)	208 (150–288)	***	0.65	**	0.11	0.69	0.07	***
Cholesterol LDL [mg/dL]	62 (42–82)	106 (61–188)	95 (61–138)	106 (64–170)	126 (89–188)	***	0.41	**	0.20	***	***	***
Cholesterol HDL [mg/dL]	74 (34–99)	57 (33–90)	52 (33–63)	54 (35 –90)	61 (40–83)	**	0.85	0.38	0.81	**	*	0.35
Triglycerides [mg/dL]	78 (37–98)	73 (34–294)	66 (34–294)	76 (41–128)	100 (37–269)	0.29						
TSH [mIU/L]	Not tested	1.9 (0.6–5.1)	1.6 (0.6–3.9)	2.2 (1.1–5.1)	1.5 (0.6–3.5)	0.16						
fT4 [pmol/L]	Not tested	12.6 (9–15)	12.8 (11.3–15)	12.3 (9.0–14.7)	12.6 (10.1–14.9)	0.55						
**Anti-inflammatory cytokines**												
IL—35 [ng/mL]	6.9 (1.2–15.6)	4.1 (1.2–22.7)	11.9 (3.9–22.7)	3.3 (1.4 –12.2)	3.3 (1.2–13.4)	***	***	***	0.99	***	0.27	0.20
IL—4 [pg/mL]	2.5 (0–6)	6.6 (0–29)	14.2 (2.2–29)	4.1 (0–21.8)	4.8 (1.3–22.9)	***	*	0.07	1	***	0.051	0.08
IL—10 [pg/mL]	2.2 (0–5.7)	0.9 (0–4.7)	1.8 (0.5–4.7)	0.8 (0–2.1)	0.4 (0–2.5)	***	*	**	0.88	0.76	***	***
**Pro-inflammatory cytokines**												
TNF–α [pg/mL]	0 (0–0.5)	2.4 (0–6.8)	1.6 (0–3.5)	2.6 (0–5.7)	4.3 (0–6.8)	***	0.46	***	**	**	***	***
IL—12 [pg/mL]	1.9 (0.9–4)	2.4 (0–16.3)	4.3 (0–16.3)	3.1 (0–8.1)	2.3 (0–15.7)	0.43						
IL—18 [pg/mL]	11.6 (4.7–18.3)	83 (34.6–146)	78.2 (34.6–146)	84 (44.8–122)	84 (46.6–134.5)	***	1	1	1	***	***	***
Ratio TNF–α/ IL—35	0 (0–0.07)	0.43 (0–3.44)	0.09 (0–0.43)	0.54 (0–2.92)	1.5 (0–3.44)	***	**	***		0.83	***	***

Significant results are marked with * (*p* < 0.05), ** (*p* < 0.01), or *** (*p* < 0.001). There was no significant difference between the subgroups of diabetic patients and the control group with regard to the levels of triglycerides, creatinine, and CRP, and also between the subgroups of patients with diabetes concerning the levels of TSH, fT4, and albuminuria (Table 2).

**Table 3 ijms-25-03190-t003:** Characteristics and comparison of subgroups A1 and A2. Data are presented as median (range)/mean values ± SD.

Characteristics	Cluster A		*p*
	A1 n = 16	A2 n = 15	
Males, n (%)	4 (27)	9 (56)	0.17
BMI [kg/m^2^]	20.4 (15.3–26.0)	22.6 (16.1–25.6)	0.3
Age [years]	17.1 (8.4–18.0)/ 15.7 ± 2.9	17.1 (13–17.9)/ 16.5 ± 1.6	0.65
Onset of diabetes [age]	9.1 (3.1–13.5)/9.0 ± 3.2	3.8 (1.2–7.6)/4.2 ± 2	***
Diabetes duration [years]	7.0 (1.2–9.4)/6.7 ± 2.4	12.2 (9.9–15.9)/12.3 ± 1.8	***
Insulin dose units/24 h	40 (21–70)	54 (30–100)	*
Insulin dose units/kg	0.7 (0.4–1)	0.8 (0.6–1.4)	0.10
Treatment with pump [%]	80 (0–90)	76 (0–100)	0.45
HbA_1c_ at onset of T1D [%]	7.7 (6.5–9.7)	8.15 (5.6–9.8)	0.63
Mean HbA_1c_ 1st year of T1D [%]	7.5 (6.2–8.4)	7.5 (5.8–8.6)	0.98
Mean HbA_1c_ 2nd year of T1D [%]	7.8 (5.5–10.1)	7.0 (6.2–8.6)	0.3
HbA_1c_ current [%]	6.8 (5.9–11.6)	8.6 (6.2–9.8)	0.07
Episodes of mild hypoglycemia [N/last month]	10 (0–16)	10 (0–20)	0.83
Episodes of severe hypoglycemia [N/last year]	0 (0–1)	0 (0–1)	0.74
Autoimmune thyroiditis, n [%]	5 (31)	4 (27)	0.9
Celiac disease, n [%]	4 (25)	4 (27)	0.76

Significant results are marked with * (*p* < 0.05), or *** (*p* < 0.001).

**Table 4 ijms-25-03190-t004:** Comparison of laboratory results in subgroups A1 and A2. Data are presented as median (range).

Characteristics	Cluster A		*p*
	A1 n = 16	A2 n = 15	
CRP [mg/L]	0.6 (0.2–4.8)	0.3 (0.1–1.3)	0.09
Serum creatinine [mg/dL]	0.71 (0.45–0.86)	0.77 (0.5–0.95)	0.57
Albuminuria [mg/dL]	14 (3–33)	7.1 (2.5–88)	0.28
Total cholesterol [mg/dL]	164 (129–248)	182 (127–247)	0.25
Cholesterol LDL [mg/dL]	95 (61–138)	106 (61–170)	0.3
Cholesterol HDL [mg/dL]	52 (33–63)	53 (39–75)	0.77
Triglycerides [mg/dL]	66 (34–294)	82 (45–159)	0.32
TSH [mIU/L]	1.6 (0.6–3.9)	2 (1–4.3)	0.57
fT4 [pmol/L]	12.8 (11.3–15)	12.9 (10.4–15.0)	0.83
**Anti-inflammatory cytokines**			
IL—35 [ng/mL]	11.9 (3.9–22.7)	4.7 (0.8–11.4)	***
IL—4 [pg/mL]	14.2 (2.2–29)	4.2 (0.7–17.8)	**
IL—10 [pg/mL]	1.8 (0.5–4.7)	1.6 (0–3.8)	0.47
**Pro-inflammatory cytokines**			
TNF–α [pg/mL]	1.6 (0–3.5)	1.9 (0–4.9)	0.63
IL—12 [pg/mL]	4.4 (0–16.3)	3.9 (0–9.2)	*
IL—18 [pg/mL]	78.2 (34.6–146)	77.2 (45–124.6)	0.87
Ratio TNF–α/IL–35	0.09 (0–0.43)	0.4 (0–2.97)	***

Significant results are marked with * (*p* < 0.05), ** (*p* < 0.01), or *** (*p* < 0.001).

## Data Availability

The data presented in this study are available on request from the corresponding author.

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
