# Peer review of "The Impact of Metabolic Memory on Immune Profile in Young Patients with Uncomplicated Type 1 Diabetes"

_ijms, 2024, doi:10.3390/ijms25063190_

Round 1
Reviewer 1 Report
Comments and Suggestions for Authors
Neubauer-Geryk and colleagues have presented a manuscript on the impact of metabolic memory on inflammatory cytokines in children with T1D
The topic is an important one in pediatric endocrinology, however, there are concerns that dampened enthusiasm about the report.
1. The major issue with this report is the study design. The investigators based their design and allocation of patients on the initial A1c levels at the time of diagnosis for groups A, B, and C
However, such categorization, while reflecting their need to identify metabolic memory due to hyperglycemia, misses an opportunity to focus on stratification by partial clinical remission status, which is a more robust parameter to determine and study the concept of both hyperglycemia memory and hyperlipidemic memory. This is critical as initial hyperglycemia at diagnosis does not differentiate those who would go on to experience partial remission which is associated with better outcomes in the short and long term.
Given that the investigators have A1c values in the first 2 years of disease in their patients, they can easily derive the subjects' insulin-dose adjusted A1c levels, and use those values to differentiate the remitters from non-remitters and then do their comparisons. This will yield more robust and reproducible results.
The other concern is the wide duration of disease for inclusion, 1.2 to 12.9 years. This means that some of the subjects are still going through partial remission while others are not. These are big confounders.
2. The introduction is too long. The first paragraph, and the last two paragraphs should suffice.
3. The Discussion is also too long. It should not be more than 6 paragraphs.
4. The Method section is rather lean. More information should be provided about the study protocol.
Comments on the Quality of English LanguageThe quality of the English is good
Author Response
Neubauer-Geryk and colleagues have presented a manuscript on the impact of metabolic memory on inflammatory cytokines in children with T1D. The topic is an important one in pediatric endocrinology, however, there are concerns that dampened enthusiasm about the report.
1./ The major issue with this report is the study design. The investigators based their design and allocation of patients on the initial A1c levels at the time of diagnosis for groups A, B, and C.
However, such categorization, while reflecting their need to identify metabolic memory due to hyperglycemia, misses an opportunity to focus on stratification by partial clinical remission status, which is a more robust parameter to determine and study the concept of both hyperglycemia memory and hyperlipidemic memory. This is critical as initial hyperglycemia at diagnosis does not differentiate those who would go on to experience partial remission which is associated with better outcomes in the short and long term.
Response:
We agree with the Reviewer that initial hyperglycemia as expressed by HbA1c does not predict which individuals will experience partial remission.
It was not our aim to analyze the effect of partial remission on the current cytokine profile. In our study, we analyzed the effect of glycemic load, expressed as HbA1c, regardless of the cause (control, presence, or absence of remission) on the pro- and anti-inflammatory cytokines. A review of the literature found although cardiovascular disease risk factors in adults with diabetes are less pronounced in those with T1D compared with T2D, hyperglycemia, and its consequences seems to be the key.
Our work provides evidence that patients with poor metabolic memory have a higher incidence of pro-inflammatory processes after approximately 10 years of disease. Additional insights into lipid metabolic memory may be a valuable consideration in the early stages of type 1 diabetes in children.
2./ Given that the investigators have A1c values in the first 2 years of disease in their patients, they can easily derive the subjects' insulin-dose adjusted A1c levels, and use those values to differentiate the remitters from non-remitters and then do their comparisons. This will yield more robust and reproducible results.
Response:
We do not know the insulin doses used in these patients during the first and second year. However, after the reviewer's comments, we performed a re-analysis in which we included only patients who had diabetes for more than 2 years. This limitation did not affect the significance of the differences between the study groups. In particular, the HbA1c levels did not differ significantly in the second year of diabetes.
3./ The other concern is the wide duration of disease for inclusion, 1.2 to 12.9 years. This means that some of the subjects are still going through partial remission while others are not. These are big confounders.
Response:
Our work was dedicated to analyze the relationship between past glycemic exposure and current inflammation in young patients without microangiopathic complications. This issue does not require analysis of the impact of the occurrence of partial remission.
- /The introduction is too long. The first paragraph, and the last two paragraphs should suffice. The Discussion is also too long. It should not be more than 6 paragraphs.
Response: We have restructured the Introduction and the Discussion. We hope that both sections, in their current shorter form, are in line with Reviewer expectations.
5/. The Method section is rather lean. More information should be provided about the study protocol.
Response: Following the comments made by the reviewers, we have added the following information to the Methods section:
“The study included patients aged between 8.4 to 18 years with diabetes of more than 1.2 years who met the type 1 diabetes diagnostic criteria according to the International Society of Child and Adolescent Diabetes. Patients were divided into three groups, A1, B and C, based on their HbA1c levels at time of diagnosis and average HbA1c levels obtained during the first and second year after diagnosis., with a mean age 16.1 ± 2.3, 14.8 ± 2.2 and 14.9 ± 1.9 years, respectively. The mean age at disease onset for each group was 6.5 ± 3.6, 7.8 ± 3.4 and 9.0 ± 3.7 years, and mean diabetes duration 9.6 ± 3.5, 7.0 ± 3.9 and 5.9 ± 2.9 years, respectively. Exclusion criteria included diabetic ketoacidosis at the time of enrollment, ongoing infection, uncontrolled celiac disease, chronic kidney disease, hypothyroidism or other endocrine disorders. Patients having experienced severe hypoglycemia in the previous month were excluded. All the participants included in the healthy control group were young people with no history of chronic disease who did not take any medications.”
/.../
”The patients treated with statin therapy were not included excluded from analysis.
Severe hypoglycemia was defined as an incident of blood glucose levels <54 mg/dL within one year of the survey, but no more than one month prior to the survey, that required assistance from another person. Mild hypoglycemia was defined as an incident within one month prior to the survey with no need for assistance [52]”.
We have also added a detailed description of analytical methods to the Analysis section as follows:
“ Blood samples were collected between 7 and 9 a.m. after an overnight fast. Sera were separated from venous blood within 30 minutes and stored frozen at - 80 C for up to three months before analysis. The same blood sample was used for all measurements.
HbA1c was measured by an immunoturbidometric method using the Unimate 3 set (Hoffmann-La Roche AG, Basel, Switzerland) with a normal range of values from 3.0 to 6.0%. An enzymatic test (Roche Diagnostics GmbH, Mannheim, Germany) was used to measure fasting glucose. The level of C-reactive protein was measured by an immunochemical system (Beckman Instr. Inc., Galway, Ireland). The levels of total cholesterol, HDL cholesterol, LDL cholesterol and triglycerides were measured using Cormay enzymatic kits (Cormay, Lublin, Poland). Urinary albumin excretion was measured by an immunoturbidometric assay. Tina-quant was used (Boehringer Mannheim GmbH, Mannheim, Germany). Serum creatinine levels were measured using the CREA assay system (Boehringer Mannheim GmbH). Serum concentrations of IL-4, IL-10, IL-18, IL-12, IL-35 and TNF -α were measured by ELISA according to the manufacturer's protocol. Serum levels of IL-4, IL-10 and IL-18 were measured by immunoenzymatic ELISA (Quantikine High Sensitivity Human from R&D Systems, Minneapolis, Minn., USA) according to the manufacturer's protocol. Minimum detectable concentrations were determined by the manufacturer as 10 pg/mL, 0.5 pg/mL, and 5.15 pg/mL, respectively. Intraassay ( 2.7%, 6.6%, and 2.9%) and interassay (7.4%, 8.1%, and 8.4% ) precision performances of the assays were determined, respectively for IL-4, IL-10, and IL-18. Serum concentrations of TNFα and IL-12 were measured by ELISA (Quantikine High Sensitivity Human from R&D Systems, Minneapolis, Minn., USA) according to the manufacturer's protocol. The intra- and inter-assay coefficients for TNFα and IL-12 were 6.2%, 2.6% 2.5%, and 7.6%, respectively. Human IL-35 was measured by ELISA (Thermo Fisher Scientific, Inc., Waltham, MA, USA). Interassay CV<10% and intraassay CV<10%. Sensitivity: 9.38 pg/mL. The absorbance of IL-4, IL-10, IL-18, IL-35, IL-12 and TNF-α was read at 450 nm on a CHROMATE 4300 automated plate reader (Awareness Technology, Inc., USA). The reference curve was generated according to the manufacturer's recommendations.”
Reviewer 2 Report
Comments and Suggestions for Authors
- The title of the manuscript should be changed as “The impact of metabolic memory on immune profile in pediatric patients with uncomplicated type 1 diabetes”
- Ethical approval should be added.
- Abstract:
· Should be re-written to be more comprehensive about the highlighted results in order to sounds better.
· The phrase in the beginning of the introduction is the same phrase in the start of abstract, so write your own words please here.
- Introduction:
· It is too long and the references are not wholly relevant and so you should rewrite with more recent and relevant literatures.
- Materials and Methods:
· This section should be rewritten with more details.
· Add the references in each step because some of them are missed.
· What about the health status of the patients? And roughly the period of onset of the diabetes in each group?
· You studied age of 20, but they are not pediatric, please check this point with correlation to the title of the manuscript.
· You should highlight the all co-factors like gender, onset of the diabetes in each group, genetic history and hormonal effect on diabetes.
- Results
· Improve the resolution of all figures.
· Remove abbreviation in this section and add one separate section of abbreviation at the end of the manuscript.
· Add star in the tables instead of values of significance then said below the table that significant ……
- Discussion:
· This section should be rewritten in more organized pattern as well as more details of the discussion of the result should be highlighted with recent relevant recent references.
- Conclusion:
· Is very short, the conclusion should be rewritten in more different style than the present conclusion to give power to the manuscript.
· Line 482: remove the cited references and related data, and add your own explanation and recommendation here.
- References:
· Should be updated till 2023.
· More relevant references about the aim of the study should be added.
· Write the all authors, names not et al. in all references.
· Unify the style of references writing according to the author guide of the journal.
- Section of abbreviation at the end of the manuscript should be added.
Author Response
1/ The title of the manuscript should be changed as “The impact of metabolic memory on immune profile in pediatric patients with uncomplicated type 1 diabetes”
Response: Thanks for the Reviewer's suggestion, we have changed the title for: “The impact of metabolic memory on immune profile in young patients with Uncomplicated Type 1 diabetes”.
2/ Ethical approval should be added
Response:
Thanks for the Reviewer's suggestion. Ethical approval is included at the end of 4.1. section: “ The study was performed according to the ethical standards of the Ethical Committee of the Medical University of GdaÅ„sk and the Declaration of Helsinki of 1964, as amended, or comparable ethical standards. The Medical University of GdaÅ„sk Ethics Committee approved the study protocol (NKBBN/277/2014; NKBBN/277-512/2016). Informed consent was obtained from all participants. Parents also consented and participated with their children.” The protocol number and a statement about the Institutional Board Review were also included.
3/ Abstract: Should be rewritten to be more comprehensive about the highlighted results in order to sound better. The phrase at the beginning of the introduction is the same as at the start of the abstract, so write your own words please here.
Response:
The abstract was re-written. We hope the Abstract in its current form, is in line with Reviewer’s' expectations.
4/ Introduction
It is too long and the references are not wholly relevant and so you should rewrite with more recent and relevant literatures.
Response:
The introduction has been shortened and rewritten.
5/ Materials and Methods:
- This section should be rewritten with more details.
- Add the references in each step because some of them are missed.
- What about the health status of the patients? And roughly the period of onset of the diabetes in each group?
Response:
Following the comments made by the Reviewer, we have added the following information to the Methods section:
“The study included patients aged between 8.4 to 18 years with diabetes duration of more than 1.2 years who met the type 1 diabetes diagnostic criteria according to the International Society of Child and Adolescent Diabetes. Patients were divided into three groups, A1, B and C, based on their HbA1c levels at time of diagnosis and average HbA1c levels obtained during the first and second year after diagnosis., with a mean age 16.1 ± 2.3, 14.8 ± 2.2 and 14.9 ± 1.9 years, respectively. The mean age at disease onset for each group was 6.5 ± 3.6, 7.8 ± 3.4 and 9.0 ± 3.7 years, and mean diabetes duration 9.6 ± 3.5, 7.0 ± 3.9 and 5.9 ± 2.9 years, respectively. Exclusion criteria included diabetic ketoacidosis at the time of enrollment, ongoing infection, uncontrolled celiac disease, chronic kidney disease, hypothyroidism or other endocrine disorders. Patients having experienced severe hypoglycemia in the previous month were excluded. All the participants included in the healthy control group were young people with no history of chronic disease who did not take any medications.
/.../
”The patients treated with statin therapy were excluded from analysis.
Severe hypoglycemia was defined as an incident of blood glucose levels < 54 mg/dL within one year of the survey, but no more than one month prior to the survey, that required assistance from another person. Mild hypoglycemia was defined as an incident within one month prior to the survey with no need for assistance [52]”.
We have also added a detailed description of analytical methods to the Analysis section as follows:
“ Blood samples were collected between 7 and 9 a.m. after an overnight fast. Sera were separated from venous blood within 30 minutes and stored frozen at - 80 C for up to three months before analysis. The same blood sample was used for all measurements.
HbA1c was measured by an immunoturbidometric method using the Unimate 3 set (Hoffmann-La Roche AG, Basel, Switzerland) with a normal range of values from 3.0 to 6.0%. An enzymatic test (Roche Diagnostics GmbH, Mannheim, Germany) was used to measure fasting glucose. The level of C-reactive protein was measured by an immunochemical system (Beckman Instr. Inc., Galway, Ireland). The levels of total cholesterol, HDL cholesterol, LDL cholesterol and triglycerides were measured using Cormay enzymatic kits (Cormay, Lublin, Poland). Urinary albumin excretion was measured by an immunoturbidometric assay. Tina-quant was used (Boehringer Mannheim GmbH, Mannheim, Germany). Serum creatinine levels were measured using the CREA assay system (Boehringer Mannheim GmbH). Serum concentrations of IL-4, IL-10, IL-18, IL-12, IL-35 and TNF -α were measured by ELISA according to the manufacturer's protocol. Serum levels of IL-4, IL-10 and IL-18 were measured by immunoenzymatic ELISA (Quantikine High Sensitivity Human from R&D Systems, Minneapolis, Minn., USA) according to the manufacturer's protocol. Minimum detectable concentrations were determined by the manufacturer as 10 pg/mL, 0.5 pg/mL and 5.15 pg/mL, respectively. Intraassay ( 2.7%, 6.6% and 2.9%) and interassay (7.4%, 8.1% and 8.4% ) precision's performances of the assays were determined, respectively for IL-4, IL-10 and IL-18. Serum concentrations of TNFα and IL-12 were measured by ELISA (Quantikine High Sensitivity Human from R&D Systems, Minneapolis, Minn., USA) according to the manufacturer's protocol. The intra- and inter-assay coefficients for TNFα and IL-12 were 6.2%, 2.6% and 2.5% and 7.6%, respectively. Human IL-35 was measured by ELISA (Thermo Fisher Scientific, Inc., Waltham, MA, USA). Interassay CV<10% and intraassay CV<10%. Sensitivity: 9.38 pg/mL. The absorbance of IL-4, IL-10, IL-18, IL-35, IL-12 and TNF-α was read at 450 nm on a CHROMATE 4300 automated plate reader (Awareness Technology, Inc., USA). The reference curve was generated according to the manufacturer's recommendations.”
6/ You studied age of 20, but they are not pediatric, please check this point with correlation to the title of the manuscript.
Response:
We decided to use the entire control group in our study without any age restrictions. This group included two subjects aged 20 years. Therefore, according to the Reviewer’s suggestion, we corrected the title of the manuscript as follows: “The impact of metabolic memory on immune profile in young patients with uncomplicated type 1 diabetes”.
7/ You should highlight the all co-factors like gender, onset of the diabetes in each group, genetic history and hormonal effect on diabetes.
Response:
We have expanded the Methods section to include detailed information about the patients included and excluded from the study. We have added information about other co-existing conditions such as autoimmune thyroiditis and celiac disease in diabetic patients in both the tables and the text of the manuscript.
8/ Results
- Improve the resolution of all figures.
- Remove abbreviation in this section and add one separate section of abbreviation at the end of the manuscript.
- Add star in the tables instead of values of significance then said below the table that significant
Response
The resolution of the images and the information about the level of statistical significance have been corrected according to the recommendation of the Reviewer. In addition, we have removed abbreviations descriptions from the text of the manuscript. They have been presented as a list at the end of the manuscript.
9/ Discussion:
- This section should be rewritten in a more organized pattern as well as more details. The discussion of the result should be highlighted with recent relevant recent references.
Response:
Following the Reviewer's suggestion, we have made substantial changes to the discussion. We hope that this is the form that will be accepted by the Reviewer.
10/ Conclusion:
- Is very short, the conclusion should be rewritten in more different style than the present conclusion to give power to the manuscript.
- Line 482: remove the cited references and related data, and add your own explanation and recommendation here.
Response
We added a separate paragraph for the conclusion. It is our hope that this form will be appropriate.
11/ References:
- Should be updated till 2023.
- More relevant references about the aim of the study should be added.
- Write the all authors, names not et al. in all references.
- Unify the style of references writing according to the author guide of the journal.
- Section of abbreviation at the end of the manuscript should be added.
Response:
We would like to thank the Reviewer for noticing imperfections in the included references list.
The present manuscript is extensively revised. The reference list changed substantially due to the major revisions of the “Introduction” and “Discussion” sections. The included references are updated till 2023 and currently follow strictly the IJMS recommendation with a full list of authors and doi where possible.
According to the Reviewer's suggestion, the Abbreviations section at the end of the manuscript was included.
.
Reviewer 3 Report
Comments and Suggestions for Authors
In the paper named ” The impact of metabolic memory on pro- and anti-inflammatory cytokines in pediatric patients with uncomplicated type 1 diabetes” authors aim to investigate whether the phenomenon of metabolic memory plays a role in the immune profile of pediatric patients with uncomplicated type 1 diabetes and to analyze the relationship between their immune profile and lipid levels. Their result gives ensigns about the long-term inflammatory process. This is characterized by an imbalance between pro- and anti-inflammatory cytokines that is associated with poor glycemic control at diabetes onset and during the first year of disease duration. These findings suggest that the metabolic memory formed as a result of hyperglycemia in the early stages of T1D may modulate inflammatory pathways and lead to significant changes in the balance of pro- and anti-inflammatory cytokines.
Only minor questions are needed
1) One interesting data that must be included in the figure 4, figure 5 is the healthy group.
2) In the same way why author do not include group A2 values in table 1 and 2? Why reason for not include this group? For the diabetes onset?
3) In figure 4 and 6 the correspondence cytokine name of the black column are missing
4) Other questions are regarding to the cytokines studies. Why author do not study the IL8 and IL6 values? This values can be very interesting take into account that both are strongly related to T1D and IL6 is a multifunctional protein that has a function in chronic inflammation
Author Response
Comments on the Quality of English Language
The quality of the English is good
In the paper named ”The impact of metabolic memory on pro- and anti-inflammatory cytokines in pediatric patients with uncomplicated type 1 diabetes” authors aim to investigate whether the phenomenon of metabolic memory plays a role in the immune profile of pediatric patients with uncomplicated type 1 diabetes and to analyze the relationship between their immune profile and lipid levels. Their result gives ensigns about the long-term inflammatory process. This is characterized by an imbalance between pro- and anti-inflammatory cytokines that is associated with poor glycemic control at diabetes onset and during the first year of disease duration. These findings suggest that the metabolic memory formed as a result of hyperglycemia in the early stages of T1D may modulate inflammatory pathways and lead to significant changes in the balance of pro- and anti-inflammatory cytokines.
Only minor questions are needed
Response:
Thank you very much for your opinion and Reviewer's comments.
1./ One interesting data that must be included in Figure 4 and Figure 5 is the healthy group.
Response: According to the Reviewer's suggestion figures 4 and 5 shown in the manuscript also contain the control (H) group.
- / In the same way why author do not include group A2 values in table 1 and 2? Why reason for not include this group? For the diabetes onset?
Response:
As it was explained in the manuscript the A group was divided in half concerning the diabetes duration. The comparison of A1 and A2 allowed us to assess the effect of diabetes duration on the immune profile, separately in patients with comparable metabolic memory (Tables 3 and 4). The comparison of groups A1, B, and C, which did not differ in age, time of onset, and duration of diabetes, allowed further comparisons without significant confounding factors (Tables 1 and 2).
- / In figure 4 and 6 the correspondence cytokine name of the black column are missing
Response
On the right side, on the Y-axis of both figures, the IL-35 is described, it was placed there because of the different units.
4./ Other questions are regarding to the cytokines studies. Why author do not study the IL8 and IL6 values? This values can be very interesting take into account that both are strongly related to T1D and IL6 is a multifunctional protein that has a function in chronic inflammation.
Response
We are fully in agreement with the opinion of the Reviewer. A valuable addition to the information on the balance of the immune system would have been an IL-6 and IL-8 test. Since our immunoassay protocol was pre-specified, we did not have a chance to include these cytokines.
Round 2
Reviewer 1 Report
Comments and Suggestions for Authors
Though the authors made improvements to the manuscript, there was no change in the design of the study, as requested.
The study design and stratification should be based on a principled pathophysiological framework such as the presence or absence of the honeymoon phase of T1D, rather than initial A1c value which is influenced by several factors.
The focus on initial A1c data for stratification, and the lack of insulin dose data during the study, and the averaging of annual A1c values instead of a longitudinal representation of the 3-monthly means of the A1c values, dampened the enthusiasm for this manuscript.
Comments on the Quality of English LanguageGood
Author Response
Dear Reviewer,
In response to the review of the manuscript "The impact of metabolic memory on proand anti-inflammatory cytokines in pediatric patients with uncomplicated type 1 diabetes".
First of all, we would like to thank the Reviewer for their efforts in reviewing our work.
Despite our belief that the information included in the Introduction and Discussion Sections
of the original manuscript is of great value in terms of content and review, we accepted
Reviewer 's comment about the need to significantly reduce it. However, we do not
understand the demand made by Reviewer in subsequent responses to change the study design
to an analysis of the consequences of clinical remission on the immune profile.
“However, such categorization, while reflecting their need to identify metabolic
memory due to hyperglycemia, misses an opportunity to focus on stratification by partial
clinical remission status, which is a more robust parameter to determine and study the
concept of both hyperglycemia memory and hyperlipidemic memory. This is critical as
initial hyperglycemia at diagnosis does not differentiate those who would go on to
experience partial remission which is associated with better outcomes in the short and
long term”.
In our response to Reviewer , we emphasized that our work was dedicated to analyzing the
relationship between past glycemic exposure and current cytokine profiles in young patients
without microangiopathic complications. This issue does not require an analysis of the impact
of partial remissions in the period between disease onset and the time of the cross-sectional
study. In addition, our aim was to address the definition of metabolic memory as it has been
defined in seminal works. The DCCT study gave rise to the concept of "metabolic memory"
(1), In which hyperglycemia early in the course of disease increases the long-term risk of
complications. Meanwhile, initial normoglycemia has long-term beneficial effects in reducing
the risk of diabetes complications. Such effects persist even if later glucose levels rise. It did,
however, describe the devastating effects of elevated HbA1c levels early in the course of type
1 diabetes.
In addition to hyperglycemia-induced oxidative stress, the mechanism of metabolic memory
involves a phenomenon in which a history of hyperglycemia can induce persistent changes in
DNA methylation at key loci. It is thought that DNAme may play a role in mediating the
relationship between HbA1c and the future development of diabetes complications (2). Short
periods of exposure to high glucose levels (which often occur early in the course of disease
have been shown to increase inflammatory gene expression and oxidative stress (3) and lead
to permanent epigenetic changes (4) in cells. In animal models, retinal complications (5)
persisted for 2.5 years, and increased inflammation and oxidative stress in the kidney (6)
persisted for several months despite the reversal of hyperglycemia. Metabolic memory may
be described as the induction of organ damage by prolonged exposure to hyperglycemia.
Referring to the renewed comment of Reviewer:
“Though the authors made improvements to the manuscript, there was no change in
the design of the study, as requested. The study design and stratification should be based
on a principled pathophysiological framework such as the presence or absence of the
honeymoon phase of T1D, rather than initial A1c value which is influenced by several
factors. The focus on initial A1c data for stratification, and the lack of insulin dose data
during the study, and the averaging of annual A1c values instead of a longitudinal
representation of the 3-monthly means of the A1c values, dampened the enthusiasm for
this manuscript.”
We would also like to emphasize that our conclusions are not based on stratification by initial
HbA1c, but on the analysis of subgroups divided based on algorithms that aggregate HbA1c
into one of three variables (HbA1c: initial, 1-year average, 2-year average) using minimization
of Euclidean distances between data (SAS, FASTCLUS procedure). Despite the concerns
raised by Reviewer, the comparison between these groups, which did not differ in age or
disease duration, allowed us to demonstrate significant differences in the immune profiles
related to the severity of hyperglycemia in the first years of disease.
The aim of our project was to investigate the relationship between glycemic control in the
initial disease period and the current immune profile in type 1 diabetic patients.
We showed that in young patients with type 1 diabetes without complications, a high HbA1c
in the first two years of diabetes is associated with a predominance of pro-inflammatory
cytokines and an abnormal lipid profile later in the course of the disease. We believe that
being able to directly reference our results to findings of established clinical trials (DDCT,
EDIC) makes our work even more valuable.
We hope that the above explanations justify our disagreement with Reviewer 's objection to
the form of presented work.
On behalf of the authors
Sincerely
Jolanta Neubauer-Geryk
References
1. Writing Team for the Diabetes Control and Complications Trial/Epidemiology of Diabetes
Interventions and Complications Research Group. Sustained effect of intensive treatment of type 1
diabetes mellitus on development and progression of diabetic nephropathy: the Epidemiology of
Diabetes Interventions and Complications (EDIC) study. JAMA 2003;290:2159–2167
2. Chen Z, Miao F, Braffett BH, et al.; DCCT/ EDIC Study Group. DNA methylation mediates
development of HbA1c-associated complications in type 1 diabetes. Nat Metab 2020;2:744–762
3. Li SL, Reddy MA, Cai Q, et al. Enhanced proatherogenic responses in macrophages and vascular
smooth muscle cells derived from diabetic db/db mice. Diabetes 2006;55:2611–2619 24.
4. El-Osta A, Brasacchio D, Yao D, et al. Transient high glucose causes persistent epigenetic changes
and altered gene expression during subsequent normoglycemia. J Exp Med 2008;205:2409–2417 25.
5. Engerman RL, Kern TS. Progression of incipient diabetic retinopathy during good glycemic control.
Diabetes 1987;36:808–812
6. Kowluru RA, Abbas SN, Odenbach S. Reversal of hyperglycemia and diabetic nephropathy: effect of
reinstitution of good metabolic control on oxidative stress in the kidney of diabetic rats. J Diabetes
Complications 2004;18:282–288

Round 3
Reviewer 1 Report
Comments and Suggestions for Authors
My prior comments on study design were not addressed
Comments on the Quality of English LanguageOK